

# A new finite element approach to model microscale strain localization within olivine aggregates

Furstoss Jean[1,2], Petit Carole[1], Ganino Clément[1], Bernacki Marc[2], and Pino-Muñoz Daniel[2]

[1]Université Nice Côte d'Azur, CNRS, OCA, IRD, Géoazur, France
[2]MINES ParisTech, PSL Research University, CEMEF-Centre de mise en forme des matériaux, CNRS UMR 7635, France

**Correspondence:** Furstoss Jean, now at Université de Lille : jean.furstoss@univ-lille.fr

**Abstract.**

This paper presents a new mesoscopic full field approach for the modelling of microstructural evolutions and mechanical behavior of olivine aggregates. The mechanical framework is based on a reduced crystal plasticity (CP) formulation which is adapted to account for non-dislocation glide strain-accommodating mechanisms in olivine polycrystals. This mechanical

description is coupled with a mixed velocity/pressure finite element (FE) formulation through a classical crystal plasticity finite element method (CPFEM) approach. The microstructural evolutions, such as grain boundary migration and dynamic recrystallization, are also computed within a FE framework using an implicit description of the polycrystal through the level-set approach.

This numerical framework is used to study the strain localization, at the polycrystal scale, on different types of pre-existing

shear zones for thermomechanical conditions relevant to laboratory experiments. We show that both fine-grained and crystallographic textured pre-existing bands favor strain localization at the sample scale. The combination of both processes has a large effect on strain localization, which emphasizes the importance of these two microstructural characteristics (texture and grain size) on the mechanical behavior of the aggregate.

## 1  Introduction

Strain localization is of first importance in the development and evolution of a tectonic plate regime (Tommasi et al. , 2009). Indeed, plate boundaries are rheological weak zones, localizing strain and allowing for tectonic plates to move with respect to each other as almost rigid blocs.

At the plate scale, the rheological behavior is mainly controlled by the upper mantle (Heron et al. , 2016) which is the strongest layer composing the lithosphere. The strain localizing mechanisms are thus often studied regarding to the upper mantle rocks,

which are mainly composed of olivine. Once localization is initiated, the positive feedback between deformation and shear heating leads to strong rheological weakening (Duretz et al. , 2019). However, the microscale processes initiating this localization are still debated. Two main mechanisms are often invoked : the plastic anisotropy of olivine (Tommasi et al. , 2009) and the

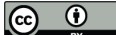

relatively weaker behavior of fine grained olivine polycrystals compared to coarse grained aggregates when deformed under grain size sensitive (GSS) creep (Braun et al. , 1999). Those two weakening processes are supported by several experimen-

tal works on olivine aggregates deformation showing that both the development of strong olivine lattice preferred orientation (LPO) (Hansen et al. , 2012) and grain size reduction (Drury , 2005) severely reduce the rheological behavior of the aggregate and may cause strain localization. Shape preferred orientation (SPO) has also been proposed as a factor involving viscous anisotropy during olivine aggregate diffusion creep (Wheeler , 2010), which potentially implies strain localization. However, the weakening intensity due to each of these processes is not straightforward to quantify experimentally.

Regarding micromechanical models, different approaches can be found in the state of the art, from 2D finite element (FE) viscous models accounting for grain size reduction and growth via front tracking methods (Jessel et al. , 2005), to 3D Fast Fourier Transform (FFT) methods accounting for crystallographic orientations through a crystal plasticity (CP) - like framework without modeling explicitly microstructural evolutions (Castelnau et al. , 2008).

Here we present a global FE framework describing the mechanical behavior of deforming olivine aggregates by a CP model

in which we incorporate a relaxation mechanism representing non-dislocation glide strain-accommodating mechanisms. The microstructural evolutions, including grain boundary migration (GBM) driven by capillarity and stored energy, solute drag and nucleation, are computed within the FE framework through the level-set (LS) approach (Scholtes et al. , 2016, 2015; Furstoss et al. , 2018). This whole numerical framework constitutes a global tool for the modeling of olivine polycrystal deformation and also allows discriminating between creep regimes controlled by grain interiors or by grain boundaries. This approach is

then used to quantify the intensity of strain localization on a pre-existing shear zone. Different type of pre-existing shear zones are tested and compared to each other, which brings new insights on the relative importance of microscale localizing features on the large-scale mechanical behavior of a grain aggregate.

## 2  Full field modeling of olivine deformation and microstructural evolutions

In this section, we present the numerical framework used to model the olivine aggregate creep and the subsequent microstruc-

tural evolutions. First we present the CP model used to model dislocation creep of olivine. Then we show that a reduced crystal plasticity formulation could be suitable to account for other strain-accommodating mechanisms in olivine aggregates. Finally we introduce the LS approach used to model microstructural evolutions such as grain boundary migration (GBM) and nucleation.

The succession of the different numerical operations described in the following is summarized by a flow chart presented in

Fig.1.



## 2.1 Crystal plasticity

CP models generally propose to decompose the plastic deformation into a combination of contributions coming from the crystal slip systems. CP calculations allow computing a mechanical behavior at a scale smaller than the grain scale and larger than the dislocation scale. Thus, in order to describe the mechanical behavior of a polycrystal, an homogenization scheme has to be

used in combination with the CP computations. The homogenization technique used in this work is the crystal plasticity finite element method (CPFEM) in which each element of the FE mesh is considered as a crystallite (Sarrazola et al. , 2020). The CP equations are then solved on each element of the mesh between each FE resolution. The P1+/P1 mixed velocity/pressure finite element (FE) formulation used for the CPFEM homogenization permits to deal with elastic anisotropy and is presented in details in (Furstoss et al. , 2021).

### 2.1.1 Numerical framework

The single CP model used here describes the stress-strain behavior of a material point within the crystal (material point crystal plasticity - MPCP) (Marin , 2006) through an incompressible plastic strain formulation. The linear elastic behavior of the material is described by the Hooke's law relating the elastic strain $\varepsilon_e$ (Green - Lagrange strain tensor) to the stress $\sigma$ (second Piola-Kirchhoff stress tensor) using the fourth order elastic constants matrix $C_e$ :

$$\sigma = C_e \varepsilon_e. \tag{1}$$

The plastic velocity gradient $L_p$ is expressed by a summation over the slip systems $\alpha$ through :

$$L_p = \sum_{\alpha=1}^{NbSlip} \dot{\gamma}^\alpha m^\alpha \otimes n^\alpha, \tag{2}$$

where $m^\alpha$ and $n^\alpha$ represent the slip direction and normal to glide plane respectively, $\otimes$ represent the dyadic product such as $m^\alpha \otimes n^\alpha = P^\alpha$ where $P^\alpha$ is the Schmidt tensor. The slip rate $\gamma^\alpha$ is determined using the following constitutive model

composed by a flow rule and an hardening law. The classical power-law flow rule is used here :

$$\dot{\gamma}^\alpha = \dot{\gamma}_0^\alpha |\frac{\tau^\alpha}{\tau_c^\alpha}|^{\frac{1}{m}} sign(\tau^\alpha), \tag{3}$$



where $m$ is a material parameter, $\dot{\gamma}_0^\alpha$ is the reference slip rate, $\tau^\alpha$ is the resolved stress computed using $\tau^\alpha = \boldsymbol{S} : \boldsymbol{P^\alpha}$ where : represents the double contraction and $\boldsymbol{S}$ the deviatoric stress. The critical resolved shear stresses (CRSS) $\tau_c^\alpha$ evolve during the deformation according to the hardening constitutive model. The dislocation-based hardening model considered here, first

relates the total dislocation density ($\rho^{tot} = \sum_{\alpha=1}^{NbSlip} \rho^\alpha$) to the CRSS through the Taylor equation (Taylor , 1934) :

$$\tau_c^\alpha = \tau_{c0}^\alpha + \psi \mu b^\alpha \sqrt{\rho^{tot}}, \tag{4}$$

where $\tau_{c0}^\alpha$ is the CRSS within the (almost) undeformed material, $\mu$ is the shear modulus, $b^\alpha$ is the Burgers vector norm and $\psi$ is the Taylor coefficient.

Then the hardening or softening of the material through the increase or decrease of the CRSS respectively, is computed by

changing the dislocation density according to the slip rate of the considered slip system :

$$\dot{\rho^\alpha} = (K_1 - K_2 \rho^\alpha)|\dot{\gamma^\alpha}|. \tag{5}$$

This expression is a variation of the Yoshie-Lasraoui-Jonas law (Laasraoui and Jonas , 1991) driving changes in dislocation density by an hardening parameter $K_1 (mm^{-2})$ representing the multiplication of dislocations during deformation, and a softening parameter $K_2$ representing the recovery capability of the material through dislocation annihilation for instance.

### 2.1.2   Application to olivine

The constitutive model presented above is used in this work within a CPFEM context, to model the mechanical behavior of olivine aggregate dislocation creep at 1573K.





The linear elastic behavior accounts for the orthorhombic symmetry of the olivine crystal using the following elastic constants

(in Voigt notation) (Isaak et al. , 1989) :

$$C_e(GPa) = \begin{pmatrix} 290 & 55 & 60 & 0 & 0 & 0 \\ 55 & 170 & 65 & 0 & 0 & 0 \\ 60 & 65 & 210 & 0 & 0 & 0 \\ 0 & 0 & 0 & 55 & 0 & 0 \\ 0 & 0 & 0 & 0 & 70 & 0 \\ 0 & 0 & 0 & 0 & 0 & 68 \end{pmatrix}. \tag{6}$$

The slip systems used in this work are the ones used in the literature for modelling olivine deformation (Castelnau et al. ,

2010). Indeed, only dislocations with $[100]$ and $[001]$ Burgers vectors are considered and gliding within $(100)$, $(010)$, $\{110\}$

and $(001)$, $(010)$, $(011)$, $(0\bar{1}1)$, $(021)$ planes respectively. It is worth mentioning that this list of slip systems gives only three

linearly independent slip systems, which is not sufficient to respect the Von-Mises criterion requiring five independent slip

systems to accommodate plastic deformation in any direction relative to the stress tensor.

The initial CRSS ($\tau_{c0}^{\alpha}$) are only discriminated for the slip systems with different Burgers vectors and are computed using the

dislocation dynamics computations of (Durinck et al. , 2007) which gives at 1573K, $\tau_{c0}^{[100]} = 22.7$MPa and $\tau_{c0}^{[001]} = 46.6$MPa.

The power law flow rule (eq.3) $\dot{\gamma}_0^{\alpha}$ and $m$ parameters are taken equal for all slip systems and worth $5 \cdot 10^{-3}s^{-1}$ and $0.1$

respectively.

For the Taylor relationship (eq. 4) parameters, the shear modulus is taken equal to $80$GPa and the Burgers vector norms

to $b^{[100]} = 4.76 Å$ and $b^{[001]} = 5.99 Å$. The Taylor parameter $\psi$ for olivine aggregate ranges between $0.5$ and $2.8$ (review in

(Durinck , 2005)) and we choose here to take an intermediate value of $1.1$.

Finally, the softening and hardening parameters $K_1$ and $K_2$ are calibrated to best reproduce experimental data, as explained

here below.

## 2.2 Reduced crystal plasticity

As mentioned above, the olivine slip systems do not allow to respect the Von-Mises criterion. In nature, this lack of linearly

independent slip systems is compensated by other strain-accommodating mechanisms such as dislocation climb (Gouriet et al.

, 2019), diffusion creep (Hirth and Kohlstedt , 2003) or grain boundary sliding (GBS) (Hansen et al. , 2011). Numerically, the


lack of slip systems may cause issues in the problem resolution. For instance, using CPFEM homogenization scheme with an

unclosed flow surface implies that the elements in which no slip system can be activated will deform elastically. The stress

gradients between these elements and neighboring elements where plasticity is active are considerable and unrealistic and the

FE resolution becomes cumbersome or even impossible. This issue is generally bypassed in some models of olivine plastic

deformation by adding some artificial slip systems permitting to satisfy the Von-Mises criterion (Tommasi et al. , 2009). These

dummy slip systems are added with very high CRSS and do not participate to the lattice rotation, which permits to model the

lattice preferred orientation (LPO) induced by dislocation glide only. Another alternative is to introduce a relaxation mechanism

accommodating strain within the deformation space not covered by the slip systems. This has been made for the modeling of

olivine deformation within a visco-plastic self consistent (VPSC) framework (Detrez et al. , 2015) and could be done within

CPFEM context by using a reduced crystal plasticity approach (Maresca et al. , 2016).

### 2.2.1  Numerical framework

The main idea of the reduced CP is to decompose the plastic velocity gradient into a part coming from dislocation glide on the

defined slip systems (as in eq.2) and a part representing the strain accommodated by a relaxation mechanism $\boldsymbol{L_{rel}}$ (Maresca et

al. , 2016) :

$$\boldsymbol{L_p} = \boldsymbol{L_{rel}} + \sum_{\alpha=1}^{NbSlip} \dot{\gamma}^{\alpha} \boldsymbol{P^{\alpha}}. \tag{7}$$

The computation of $\boldsymbol{L_{rel}}$ requires to define the part of the deformation space not covered by the slip systems. For this, the

list of the different Schmidt tensors has to be expressed as a base composed by a set of linearly independent and orthonormal

tensors $\boldsymbol{M^{\beta}}$ such as (Maresca et al. , 2016) :

$$\sum_{\alpha=1}^{ns} \lambda^{\alpha} \boldsymbol{P^{\alpha}} = \sum_{\beta=1}^{N} \mu^{\beta} \boldsymbol{M^{\beta}}, \tag{8}$$

where $\lambda^{\alpha}$ and $\mu^{\beta}$ are non null real. It could be noticed that if all the slip systems are linearly independent $N = ns$ (in a general

case $N > ns$). In fact, the base formed by these tensors represents the part of the deformation space in which the strain can be





accommodated by dislocation glide. Thus, one can define a fourth order projector $\mathbb{P}$ allowing to project any stress tensor within

the deformation space not spanned by the slip systems as (Maresca et al. , 2016) :

$$\mathbb{P} = \mathbb{K} - \sum_{\beta=1}^{N} (\boldsymbol{M^\beta})^{\boldsymbol{T}} \otimes (\boldsymbol{M^\beta}), \tag{9}$$

where $\mathbb{K}$ is the fourth order identity tensor. It only remains to define the strain accommodated by the relaxation mechanism $\varepsilon^{rel}$

in response to the projected deviatoric stress $\boldsymbol{S^{\mathbb{P}}} = \boldsymbol{S} : \mathbb{P}$ (Maresca et al. , 2016) :

$$\boldsymbol{L_{rel}} = \dot{\varepsilon}^{rel} \frac{3}{2S_{eq}} \boldsymbol{S^{\mathbb{P}}}, \tag{10}$$

where $\dot{\varepsilon}^{rel}$ is the plastic strain rate of the relaxation mechanism and :

$$S_{eq} = (\frac{3}{2} \boldsymbol{S^{\mathbb{P}}} : \boldsymbol{S^{\mathbb{P}}})^{\frac{1}{2}}. \tag{11}$$

The simplest way to express the relaxation plastic strain rate is to take, similarly to the power law flow rule (eq.3), a Von-

Mises type isotropic expression such as (Maresca et al. , 2016) :

$$\dot{\varepsilon}^{rel} = \dot{\varepsilon}^{iso} = \dot{\varepsilon}_0 (\frac{S_{eq}}{\sigma_c^{iso}})^{\frac{1}{n}}, \tag{12}$$

where $\dot{\varepsilon}_0$ and $n$ are two parameters and $\sigma_c^{iso}$ is a critical stress for the activation of the relaxation mechanism (similarly to the

CRSS).

As for the hardening or softening of the dislocation glide presented above, the strength of the relaxation mechanism could also

evolve during the deformation. However this aspect will not be taken into account in this work (i.e. $\dot{\sigma}_c^{iso} = 0$).

It is worth mentioning that the relaxation mechanism, as formulated and introduced above, is not involved in the lattice ro-

tation computation. Thus, the introduction of this mechanism should not impact directly the development of LPO. The only





influence on the lattice rotation may be indirect, by minimizing the elastic rotation through the slightest activation of the elastic deformation regime.

### 2.2.2 Application to olivine

In this work, the relaxation mechanism introduced by the reduced CP presented above is used to model the strain-accommodating processes taking place in olivine aggregates, excluding dislocation glide which is accounted for in the classical CP (section 2.1). Two main families of strain-accommodating mechanisms are distinguished here, the ones taking place at grain interiors and the ones occurring at grain boundaries (GB). The first ones combine the effect of dislocation climb (Gouriet et al. , 2019) and diffusion creep at grain interiors (i.e. Nabarro-Herring creep (Nabarro , 1948; Herring , 1950)) while the latter should

represent mechanisms such as GB sliding (GBS) (Hansen et al. , 2011) and its variations (dislocation accommodated GBS (Wang et al. , 2010), superplasticity (Hiraga et al. , 2010)) and diffusion creep within grain boundaries (i.e. Coble creep (Coble , 1963)). Even if numerical solutions exist for the modelling of dislocation climb (Geers et al. , 2014) and GBS (Wheeler , 2010), the use of a unique CPFEM framework is more suitable and numerically efficient (as CPFEM is already numerically expensive). Thus, the parameters of the relaxation mechanism in its isotropic formulation (Eq.12) should reflect the relative

importance on the aggregate deformation between the strain-accommodating mechanisms and dislocation glide. As we choose to take the $\varepsilon_0$ and $n$ parameters constant and equal to the parameters of the dislocation glide power law (Eq.3) $\dot{\gamma}_0^\alpha$ and $m$, the only parameter we can use to discriminate between grain boundaries and interiors is the relaxation strength $\sigma_c^{iso}$.

For grain interiors, we choose a $\sigma_c^{iso,GI}$ as small as possible to left unchanged the mechanical response of the monocrystal deformation simulations used to calibrate the hardening and recovery parameters $K_1$ and $K_2$ (see section 3.1.1), we get

$\sigma_c^{iso,GI} = 20\tau_{c0}^{[001]}$.

Recent laboratory observations report an amorphization olivine around grain boundaries due to the application of stress (Samae et al. , 2021). An hypothesis on the origin of strain accommodating mechanisms at grain boundaries in olivine may be related to this amorphization. The stress needed to amorphize a crystal can be seen, at first order, as the stress above which the lattice is no more stable. Ab initio calculations make it possible to compute an ideal shear/tensile stress (ISS/ITS) representing a critical

stress, at 0K, above which the structure of a perfect crystal is no more stable and could change to another phase, which could be amorphous. The ITS/ISS computed for olivine (Gouriet et al. , 2019) show magnitudes relatively low and similar to the ones of Peierls stresses (Durinck , 2005) almost corresponding to the CRSS at 0K. Thus, in this work we consider the strength of the relaxation mechanism at grain boundaries $\sigma_c^{iso,GB} = \tau_{c0}^{[001]}$.





The width around interfaces in which the relaxation mechanism is considered as the one representing GBS (or other strain

accommodating mechanisms at GB) will be called the mechanical GB width in the following. It could be noticed that this

width does not correspond to the chemical or physical GB width. In fact, this width should represent the zone around GB

in which the mechanical behavior of the crystal is impacted by the presence of the boundary. Such a width has never been

described before in olivine but its value can be taken close to the grain size below which the creep regime of the aggregate is

controlled by GB. In the thermomechanical conditions considered in this work (1573K and strain rate of $10^{-5}s^{-1}$) this value

is close to $10\mu m$ (Skemer et al. , 2011). Thus the mechanical GB width will be taken to 4 elements with a mean mesh size of

$3\mu m$.

Using the slip systems presented in section 2.1.2, the linearly independent and orthonormal tensors $M^{\beta}$ used to compute

the projector $\mathbb{P}$ (see Eq.9) are :

$$M^1 = \begin{pmatrix} 0 & 0 & \frac{1}{\sqrt{2}} \\ 0 & 0 & 0 \\ \frac{1}{\sqrt{2}} & 0 & 0 \end{pmatrix} \quad M^2 = \begin{pmatrix} 0 & \frac{1}{\sqrt{2}} & 0 \\ \frac{1}{\sqrt{2}} & 0 & 0 \\ 0 & 0 & 0 \end{pmatrix} \quad M^3 = \begin{pmatrix} 0 & 0 & 0 \\ 0 & 0 & \frac{1}{\sqrt{2}} \\ 0 & \frac{1}{\sqrt{2}} & 0 \end{pmatrix}. \tag{13}$$

## 2.3 Microstructural evolutions

The microstructural evolutions of the olivine aggregate during deformation are modelled using the level-set (LS) framework

(Merriman et al , 1994; Bernacki et al. , 2011; Furstoss et al. , 2020). This formalism allows describing the polycrystal using

signed distance functions (LS functions) representing the distance to the GB and are defined positive inside the grain they

represent and negative elsewhere. In order to reduce the number of LS functions needed to describe the aggregate, we use

global LS (GLS) functions representing several non-neighboring grains (Scholtes et al. , 2015). The displacement of the GLS

are computed within a 3D FE framework using an isotropic mesh composed of P1 tetrahedral elements with a typical mean size

of $3\mu m$. After 10% of macroscopic deformation we use isotropic homogeneous remeshing in order to get elements with good

quality. The initial polycrystals are generated using an optimized dropping and rolling method and a Laguerre-Voronoï tesse-

lation algorithm allowing us to respect a precise grain size distribution (Hitti et al. , 2012).





### 2.3.1 Grain boundary migration


Modelling GBM within a LS framework is equivalent to advect the GLS functions according to the equations of GBM. In this work we use a capillarity and stored energy-driven GBM accounting for solute drag such as the grain boundary velocity $\boldsymbol{v}$ can be expressed as :

$$\boldsymbol{v} = M(-\gamma\kappa + \tau\Delta\rho - P_d)\boldsymbol{n}, \tag{14}$$

where $\boldsymbol{n}$ and $\kappa$ are the grain boundary outward unit normal and sum of main curvatures respectively, $M$ and $\gamma$ are the olivine/olivine GB mobility and interfacial energy respectively. The latters are considered as constant and are taken as $2.9 \cdot 10^{-2} mm^4.J^{-1}.s^{-1}$ and $1 \cdot 10^{-6} J.mm^{-2}$ respectively according to (Furstoss et al. , 2018). $\tau = 1.3 \cdot 10^{-11} J.mm^{-1}$ (Durinck , 2005) and $\rho$ are the dislocation line energy and density, respectively. $P_d$ is the solute drag pressure which can be approximated as :

$$P_d = \frac{\alpha c_0 |\boldsymbol{v}|}{1 + v^2 \beta^2}, \tag{15}$$

where $c_0$ is the impurity concentration taken as 200ppm, $\alpha$ and $\beta$ are the solute drag parameters and are taken as $3 \cdot 10^8 J.s.mm^{-4}$ and $4.8 \cdot 10^8 s.mm^{-1}$, respectively, according to (Furstoss et al. , 2020).

The FE strong formulation for the displacement of the GLS functions $\Phi_i$ is obtained using the same semi-explicit first order time discretization presented in (Furstoss et al. , 2020) in addition with the accounting of stored energy :

$$\begin{cases} \mathcal{M}\frac{\Phi_i^{t+\Delta t}}{\Delta t} - M\gamma\Delta\Phi_i^{t+\Delta t} + \boldsymbol{v_e}\boldsymbol{\nabla}\Phi_i^{t+\Delta t} = \mathcal{M}\frac{\Phi_i^t}{\Delta t}, \\ \boldsymbol{v_e} = M\tau\Delta\rho\boldsymbol{n}, \end{cases} \tag{16}$$

where $\Delta t$ is the timestep and $\mathcal{M}$ is a term (equivalent to a mass term by analogy with the heat equation) permitting to account for the solute drag effect (Furstoss et al. , 2020). The difference of dislocation density across GB $\Delta\rho$ is computed by averaging the dislocation density per interface following the methodology presented in (Ilin et al. , 2018). The boundary conditions



applied to GLS functions are null Neumann ones (i.e. $\boldsymbol{\nabla}\Phi \cdot \boldsymbol{n}_\Gamma = 0$ with $\boldsymbol{n}_\Gamma$ the unit normal to domain) which imposes to GLS

functions to evolve towards a direction normal to the calculation domain boundaries.

After each GLS transport step, a reinitialization procedure is applied to the GLS functions in order to restore their metric properties (i.e. $||\boldsymbol{\nabla}\Phi|| = 1$) at least in a thin layer around interfaces. This reinitialization step is performed using the direct approach proposed by (Shakoor et al. , 2015).

The volume swapped during GBM is tracked and we assign to it a low dislocation density $\rho_0 = 10^5 mm^{-2}$ corresponding to

the non-hardened material. The lattice orientation in the elements swapped during GBM is defined as the orientation of the neighboring element belonging to the same grain and having the minimal dislocation density (i.e. the less deformed one).

### 2.3.2   Dynamic recrystallization

Olivine aggregates evidence mainly continuous dynamic recrystallization (DRX) (Poirier and Nicolas , 1975) through sub-grain rotation (SGR) mechanism. In the LS framework used here, only the discontinuous nucleation (DDRX) can be considered by

inserting nuclei and make them growth. The nucleation is accounted for within the numerical framework by simply modify some well-chosen GLS functions, or if no GLS function can receive the nucleus, by adding a new GLS function (Scholtes et al. , 2016). Potential nucleation sites are not restricted at zones around GB but can be anywhere in the whole aggregate. In order to account for the continuous nature of the DRX in olivine, we propose to model the SGR process by considering each nucleus as a sub-grain surrounded by low angle GB (LAGB). As LAGB generally have a much lower mobility than high angle

grain boundaries (HAGB) (Humphreys and Hatherly , 2012) we initially ascribe them $M^{LAGB} < M^{HAGB}$. In nature, as the sub-grain deforms it will progressively rotate from the parent grain and the mobility of the surrounded GB will reach the one of LAGB (Humphreys and Hatherly , 2012). This mechanism is accounted for in our numerical framework by defining the mobility for the GB surrounding a nucleus as a function of the strain accumulated within the nucleus such as :

$$M(\bar{\varepsilon}_g^{cum}) = \begin{cases} M^{LAGB}\left(\dfrac{M^{HAGB}}{M^{LAGB}}\right)^{\frac{\bar{\varepsilon}_g^{cum}}{\bar{\varepsilon}_{HAGB}^{cum}}} & \text{if} \quad \bar{\varepsilon}_g^{cum} \leq \bar{\varepsilon}_{HAGB}^{cum}, \\ M^{HAGB} & \text{if} \quad \bar{\varepsilon}_g^{cum} > \bar{\varepsilon}_{HAGB}^{cum}, \end{cases} \tag{17}$$

where $\bar{\varepsilon}_g^{cum}$ is the average strain accumulated within the nucleus and $\bar{\varepsilon}_{HAGB}^{cum}$ is the maximal accumulated strain before the sub-grain becomes a new grain.

Nucleation criteria are generally defined as either large orientation or dislocation density gradients or high dissipated energy or dislocation density. In the present work the nucleation criterion is only based on a critical dislocation density $\rho^{cr}$ above





which the nucleation probability is non-null. This critical dislocation density is computed in a way permitting to respect the

Bailey-Hirsch criterion which relate the critical radius for a spherical nuclei $R^{Nucl}$ to balance out the capillarity force by the

force due to the difference of stored energy :

$$\rho^{cr} = \omega \frac{2\gamma}{R^{Nucl}\tau}, \tag{18}$$

where $\omega$ is a security factor taken as 2.

The nucleus radius is computed following the piezometric relationship defined by (Van der Wal et al. , 1993) :

$\sigma(MPa) = 15.10^2 R^{Nucl\,-0.8}(\mu m), \tag{19}$

in combination with the diffusion creep law (Hirth and Kohlstedt , 2003) in which the grain size is replaced by the nucleus size

:

$$\dot{\varepsilon} = A\sigma R^{Nucl\,-3}e^{-\frac{E}{RT}}, \tag{20}$$

where $R$ is the perfect gas constant, $T$ is the temperature, $\dot{\varepsilon}$ is the strain rate, $A = 1.5 MPa^{-1}.mm^3.s^{-1}$ and $E = 375 kJ.mol^{-1}$.

Combining Eq. 19 and 20 gives the nucleus radius as a function of the strain rate (that is imposed in our simulation) and Eq.

18 gives the critical dislocation density.

The nucleation rate $\dot{V}$, representing a nuclei volume per time unit, is computed using the following expression (Maire et al.

, 2017) :

$$\dot{V} = K_g\phi\Delta t, \tag{21}$$

where $\phi$ represents the total boundary area surrounding the aggregate volume verifying $\rho > \rho^{cr}$ and $K_g(m.s^{-1})$ is a probability

coefficient which has to be calibrated with experimental data (see section 3.1.2).





As for the volume swapped during GBM, a low dislocation density $\rho_0$ is ascribed to the zones where nuclei have appeared while the nuclei orientations are defined as the initial orientations of the parent grains.

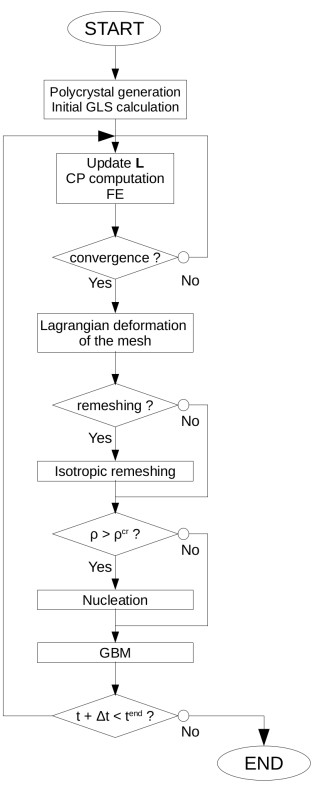

**Figure 1.** Scheme describing the different steps of a resolution increment.

## 3    Material parameters and creep regime

In this section, we present the calibration of the different material parameters used in this framework on experimental data. First, the hardening and recovery parameters $K_1$ and $K_2$ (Eq.5) are identified using a monocrystal compression test. Then we calibrate the recrystallization rate $K_g$ using experimental results on a simple shear experiment on olivine aggregates. Finally we show that our numerical framework is able to capture two different creep regimes, one in which the deformation is accommodated by the bulk material (which could be the dislocation creep) and one dominantly controlled by grain boundary

strain-accommodating mechanism.





## 3.1 Material parameters calibration

The different material parameters presented above and adjusted below are summarized in table 1.

| CP parameters | $\dot{\gamma}_0$ | $m$ | $\tau_{c0}^{[100]}$ | $\tau_{c0}^{[001]}$ | $b^{[100]}$ | $b^{[001]}$ | $\Psi$ | $K_1$ | $K_2$ |
|---|---|---|---|---|---|---|---|---|---|
| Units | s$^{-1}$ | | MPa | MPa | mm | mm | | mm$^{-2}$ | |
| Value | 5.10$^{-3}$ | 0.1 | 22.7 | 46.6 | 4.76.10$^{-7}$ | 5.99.10$^{-7}$ | 1.1 | 5.6.10$^{8}$ | 35 |
| Reduced CP parameters | $\dot{\epsilon}_0$ | $n$ | $\sigma_c^{iso,GI}$ | $\sigma_c^{iso,GB}$ | | | | | |
| Units | s$^{-1}$ | | MPa | MPa | | | | | |
| Value | 5.10$^{-3}$ | 0.1 | 932 | 46.6 | | | | | |
| Microstructural evolution parameters | $M$ | $\gamma$ | $\tau$ | $\alpha$ | $\beta$ | $C_0$ | $\rho^\sigma$ | $R^{Nucl}$ | $K_g$ |
| Units | mm$^4$.J$^{-1}$.s$^{-1}$ | J.mm$^{-2}$ | J.mm$^{-1}$ | J.s.mm$^{-4}$ | s.mm$^{-1}$ | ppm | mm$^{-2}$ | mm | mm.s$^{-1}$ |
| Value | 2.9.10$^{-2}$ | 10$^{-6}$ | 1.3.10$^{-11}$ | 3.10$^{8}$ | 4.8.10$^{8}$ | 200 | 2.68.10$^{7}$ | 1.15.10$^{-2}$ | 2.5.10$^{-6}$ |

**Table 1.** Values of the different material parameters used in this work (temperature of 1573K and strain rate of $10^{-5}s^{-1}$).

### 3.1.1 Recovery and hardening

Monocrystal deformation experiments seem the most appropriate mechanical tests to calibrate the hardening and recovery parameters $K_1$ and $K_2$ because these tests do not involve GB and thus permit to consider only the deformation accommodated by dislocation creation and motion. The experimental data selected for this calibration are the ones of (Phakey et al. , 1972) in which three different monocrystal orientations are used with respect to the compression axis. The compression tests are performed for these three orientations at 1073K and 1273K. In order to compare the strain stress curves between experiments and simulations, we use the macroscopic strain and we compute the equivalent stress $\sigma_{eq}$ as :

$$\sigma_{eq} = \frac{\int_{\Omega_{2D}} \sigma_{yy} dS}{S}, \tag{22}$$

where $\boldsymbol{y}$ is the compression direction, $\Omega_{2D}$ represents the cross section (normal to $\boldsymbol{y}$) of the sample in its middle and $S$ represents the surface of this cross section. As in experiments, we use a cylinder sample with a diameter of 3mm and a length of 4mm and a strain rate of $10^{-5}s^{-1}$. The upper face of the cylinder is moved at a constant velocity in the compression direction and the displacements in the other directions are imposed as null on this face. The movements of the lower face





in the compression direction are imposed to zero while the displacements in the other directions are left free. Finally the displacements of the other face are also left free.

The $K_1$ and $K_2$ parameters are considered only dependent on temperature and are calibrated by selecting the values permitting to best reproduce the experimental strain stress curves for the three orientations at a given temperature. The identification of the best fitting $K_1$ and $K_2$ is performed by using an optimization software, called MOOPI (Chenot et al. , 2011), based

on a genetic algorithm permitting to explore the parameter space by achieving random mutations in order to ensure that the proposed parameters do not represent a local minimum of the cost function (which represents the quality of the parameters with respect to the expected results). The strain stress curves of the best fitting hardening and recovery parameters are presented in figure 2.

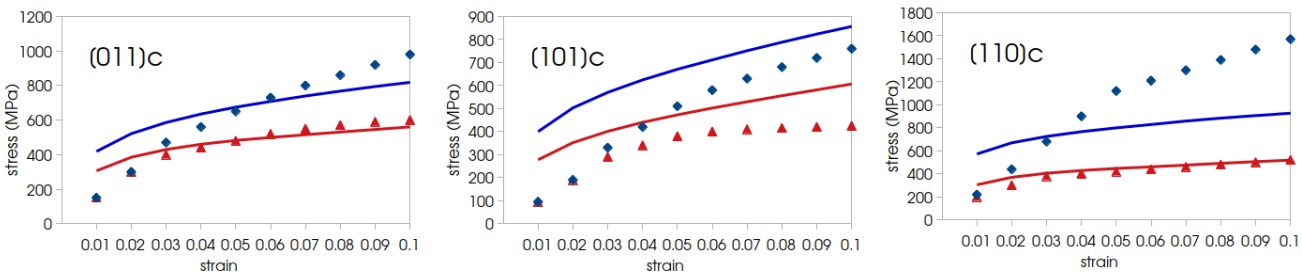

**Figure 2.** Best fitting strain stress curves for hardening and recovery parameters calibration. Solid lines represent the simulated mechanical behavior while dots represent experiments of (Phakey et al. , 1972). For the three different orientations the colors represent the temperature of the compression test, 1073K (blue) and 1273K (red).

The simulated strain stress curves show similar trends than the experimental ones excepted for the $[110]c$ test at $1073K$ in

which some experimental issues have been reported (Phakey et al. , 1972).

The values of the hardening and recovery parameters are linearly extrapolated in order to get their values at 1573K which is the temperature used in the following. At $1073K$ the best fitting $K_1$ and $K_2$ are $9.6 \cdot 10^8 mm^{-2}$ and 10 respectively, at $1273K$ they are worth $8 \cdot 10^8 mm^{-2}$ and 20 respectively, and thus at $1573K$, $K_1 = 5.6 \cdot 10^8 mm^{-2}$ and $K_2 = 35$.

### 3.1.2 Dynamic recrystallization parameters

The numerical values for the LAGB mobility $M_{LAGB}$ and the average strain accumulated above which the sub-grain (nucleus) can be considered as a grain $\bar{\varepsilon}^{cum}_{HAGB}$ (see section 2.3.2) are chosen in order for a majority of sub-grains to not disappear in the increments following nucleation. After performing several simulations we choose $M_{LAGB} = M_{HAGB}/1000$ and $\bar{\varepsilon}^{cum}_{HAGB} =$



0.3 which permits for the largest part of the sub-grains to persist at least few increments after their nucleation. Further numerical or experimental studies should be done to evaluate these parameters which are, in the current model, more numerical than

physical parameters.

For the adjustment of the dynamic recrystallization rate $K_g$, we aim at reproducing a simple shear experiment of a polycrystalline olivine performed by (Zhang et al. , 2000). For this purpose, we use a computational domain of $0.15 \times 0.15 \times 0.1mm$ initially composed by 17 grains having a mean radius size of $31.6\mu m$ (Fig.3).

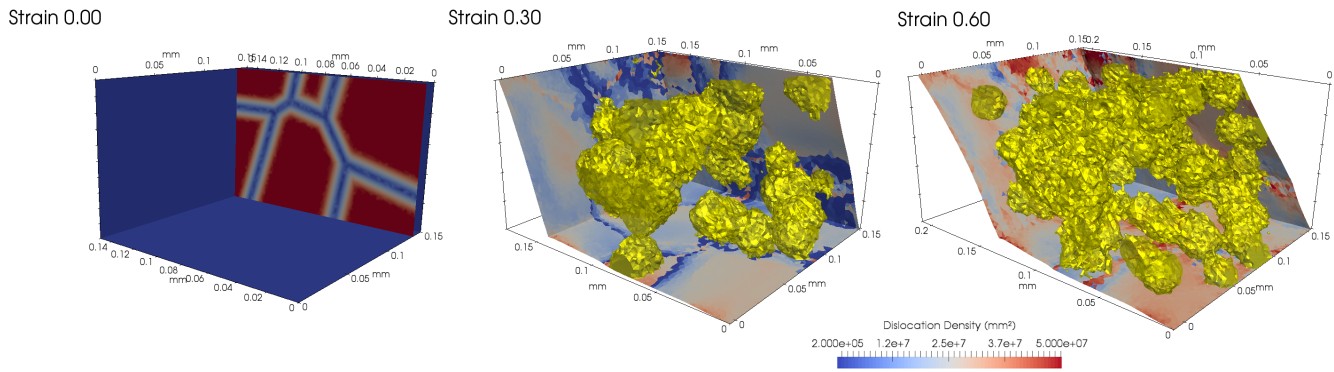

**Figure 3.** Full field simulation of olivine aggregate deformed in simple shear for the best fitting $K_g$ value and for different shear strain. Within the slice (left right picture) is represented the maximum of GLS functions which permits to visualize grain boundaries. The yellow volumes represent the dynamically recrystallized grains while the color scale represents the total dislocation density within the polycrystal.

To deform the sample in simple shear we impose null displacement in the shear direction and normal to shear plane on the

lower face of the sample while the other direction is left free. The upper face is displaced at constant velocity in the shear direction, no movements are allowed in the direction normal to the shear plane and the third direction is left free. The left and right faces are displaced in the shear direction at a constant velocity corresponding to the linear interpolation between the upper and lower face velocities in the shear direction. As for the other faces, the movements in the direction normal to shear plane are set to zero while in the other direction nothing is imposed. Finally, the movements of the front and back faces are left free

in all directions.

As in experiments (Zhang et al. , 2000), the sample is deformed at 1573K and $10^{-5}s^{-1}$. Multiple simulations with different $K_g$ values are performed in order to obtain the $20\%$ of dynamically recrystallized grain fraction after $60\%$ of shear strain measured within experiments. Using a $K_g$ value of $2.5 \cdot 10^{-9}mm.s^{-1}$ allows for having $25\%$ of dynamically recrystallized grain fraction after $60\%$ of shear strain which is the best fit obtained (Fig.3).





### 3.2 Full field modeling of olivine creep regime

In the full field framework presented above, the grain boundaries and interiors contributions to the whole deformation of the polycrystal can be followed during the mechanical test. To do this, the local equivalent strain ($\varepsilon$) can be computed from the velocity gradient using $\varepsilon = \sqrt{\frac{2}{3}\boldsymbol{L}:\boldsymbol{L}}$, and once integrated upon the whole domain $\Omega$ separated into two components such as :

$$\int_{\Omega} \varepsilon dV = \int_{\Omega_{GB}} \varepsilon dV + \int_{\Omega_{GI}} \varepsilon dV, \tag{23}$$

where $\Omega_{GB}$ and $\Omega_{GI}$ represent the part of the microstructure in which the relaxation mechanism represents the GB and grain interiors strain accommodating mechanisms respectively (see section 2.2.2).

During the deformation, DRX occurs and results in a mean grain size reduction through nucleation of new small grains. Equation 23 shows that, for mean grain size superior to $\approx 20\mu m$ the deformation is mainly accommodated in grain interiors (Fig.4).

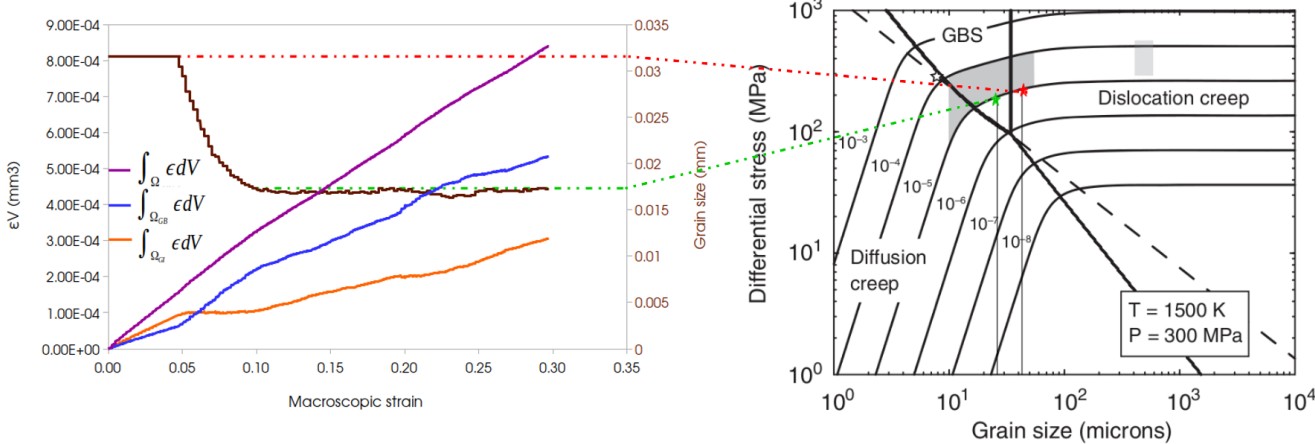

**Figure 4.** Left : equivalent strain integrated (see Eq.23) over the whole polycrystal (purple), grain boundaries (blue) and grain interiors (orange), and integrated over time. Mean grain size evolution during deformation (brown) and the corresponding thermomechanical conditions and mean grain size in an olivine deformation map (right) from (Skemer et al. , 2011).

As strain increases, mean grain size decreases under $20\mu m$ and deformation of the aggregate becomes dominated by deformation mechanisms occurring at GB (Fig.4). The transitional grain size between GB and grain interior controlled creep determined



from this analysis is in accordance with deformation maps of olivine (Skemer et al. , 2011) obtained from laboratory experiments (Fig4).

These observations allow us to propose that this framework is at least capable to discriminate two different creep regimes, one monitored by GB strain-accommodating mechanism and one mainly controlled by grain interiors.

It is worth noticing that this transition is mainly due to the increase of the volumetric fraction of GB with the decrease of mean grain size due to DRX. Indeed, the mechanical grain boundary width chosen for this study (see section 2.2.2) implies that for an aggregate with grain radii inferior to $\approx 10 \mu m$, the majority of the aggregate behaves as GB.

## 4 Results

In the following section, the presented numerical framework is used to study the strain localization on different sorts of pre-existing microscale shear zones. For this we use computational domains of $0.69 \times 0.25 \times 0.35 mm$ deformed in traction and we adopt a pre-existing shear band geometry similar to one used in (Tommasi et al. , 2009) i.e., a shear band oriented at $45°$ with respect to the extension direction (Fig.5).

The left face of the domain is displaced at a constant velocity in the traction direction imposing a macroscopic strain rate of $10^{-5} s^{-1}$ while the displacements in the other directions are imposed to be null. The right face cannot move in the traction direction but is free in the other ones. The movements of the other faces are left free in all directions.

The simulations are performed at a temperature of $1573K$ and using the parameters described before (see table 1).

### 4.1 Initial setups

Five different models are tested here, one with no pre-existing shear zone which will serve as a reference case and four others with different types of pre-existing shear bands.

The first considered shear zone consists in a strongly textured (with large lattice preferred orientation) zone in which grains have orientations maximizing the dislocation plastic strain rate (Eq.2) while outside of the shear zone, grains have random orientations (Fig.5).

The second shear zone corresponds to an ultrafine grained zone in which the mean grain size is $\approx 10 \mu m$, while elsewhere the mean grain size is of $\approx 30 \mu m$. In the rest of the polycrystal, grain orientations are picked randomly (Fig.5).

The third one is composed of strongly elongated grains where the maximal elongation direction is the shear zone direction ($45°$ to the extension direction). In this setup, called SPO (shape preferred orientation) shear zone in the following, the grain orientations are randomly selected (Fig.5).



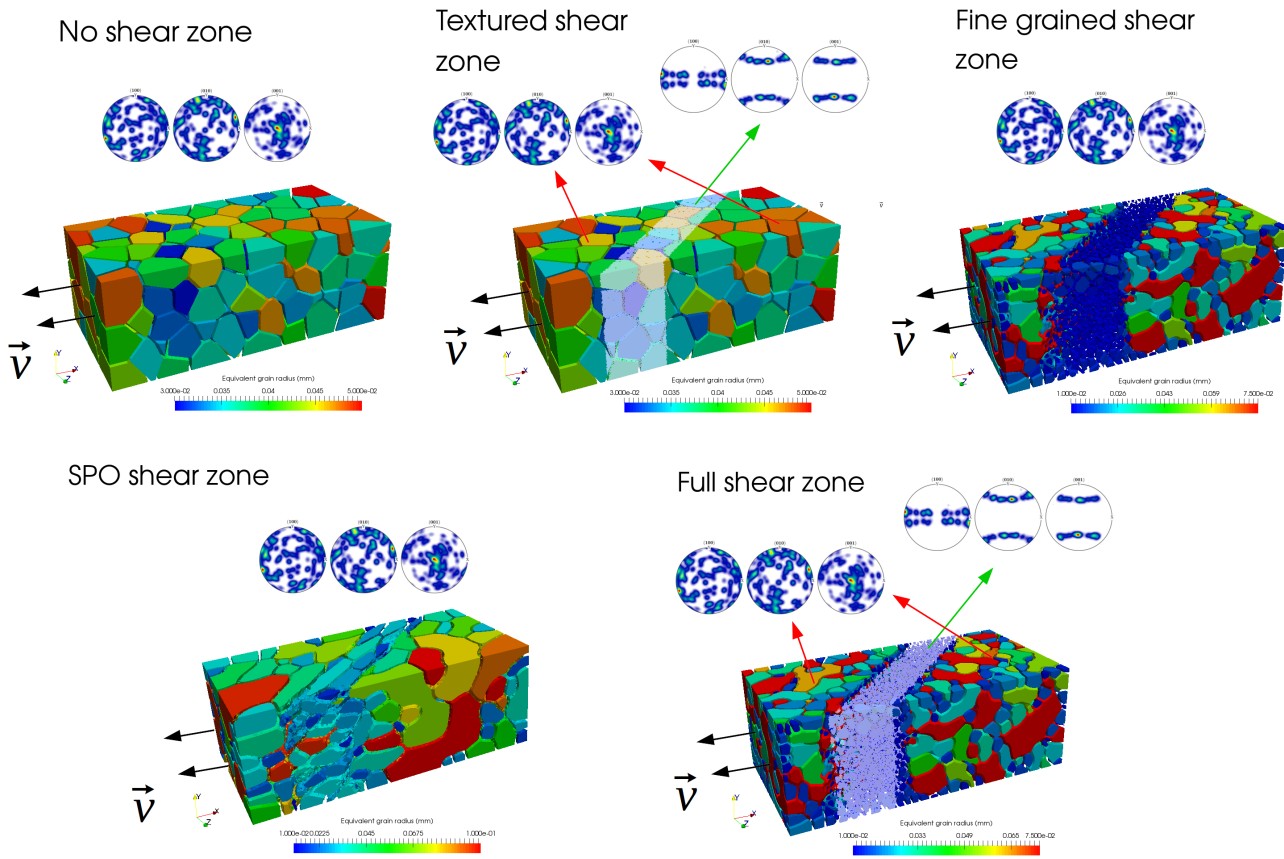

**Figure 5.** Initial setups of the performed simulations, color scale represent equivalent grain radius. Orientations within the different zones are represented through pole figures computed using MTEX (Mainprice et al. , 2015).

The last shear zone consists in a combination of the two first ones such as the zone is composed of ultrafine grain with

orientations maximizing the dislocation plastic strain rate. Elsewhere, the grain size are $\approx 30 \mu m$ and their orientations are randomly selected (Fig.5).

### 4.2   Full field results

The efficiency of strain localization in the numerical sample can be examined by comparing the amount of strain accumulated inside and outside of the shear band between the different pre-textured models and the reference model (Fig.6).

For the case in which no specific shear zone has been imposed, the inner part of the domain accommodates more strain than the outer parts ($\approx$20% higher after 8% of macroscopic strain). In this case, the larger strain rates are mainly observed at GB





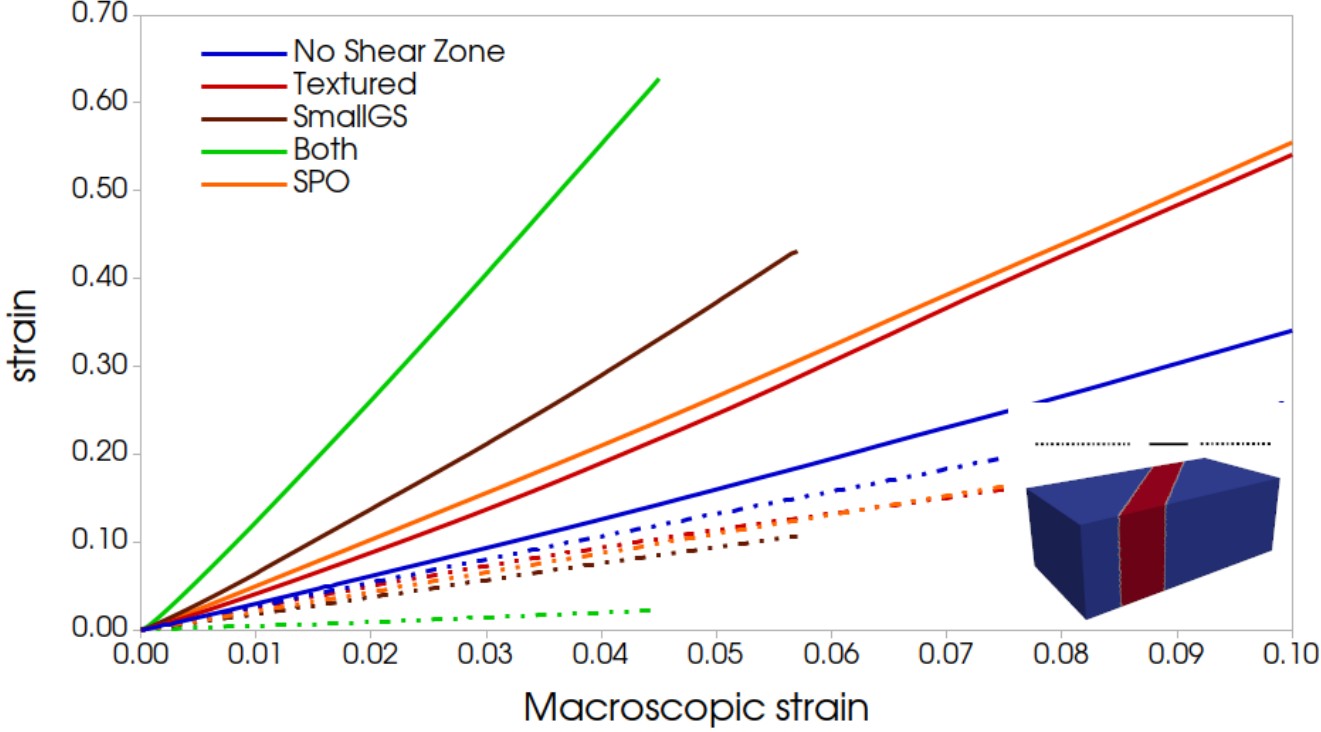

**Figure 6.** Cumulated average strain for the inner (solid lines) and outer (dashed lines) parts of the sample for the different type of pre-existing shear zone.

in the whole microstructure and DRX occurs in the sample where dislocation density is the highest (Fig.7a). For the textured shear zone case, strain localization in the inner zone is much more pronounced (110% higher than the outer parts after 8% of macroscopic strain, Fig.6). Moreover, DRX through nucleation events occurs preferentially within the shear zone as well as on

the outskirts of the shear zone (Fig.7b). Here again, large strain rates seem to localize at GB, and more precisely at the interface between well oriented (for dislocation glide) and badly oriented grains. For the fine grained shear zone, the inner part of the sample accumulates the largest part of the deformation (140% higher than the outer parts after 6% of macroscopic strain, Fig.6). A large, continuous and quickly deforming shear band (with high strain rate) is observed in the inner zone while nucleation also occurs preferentially around and within the shear zone (Fig.7d). For the SPO model, the partition of deformation between

the inner and the outer zones is nearly equivalent to that of the textured shear zone model (Fig.6). The strain rate distribution is different from the fine grain case : in fact we can observe the formation of multiple shear bands parallel to the initial shear zone and located around this area (Fig.7c). As for other models, the DRX events are mainly located near the shear zone. For the shear



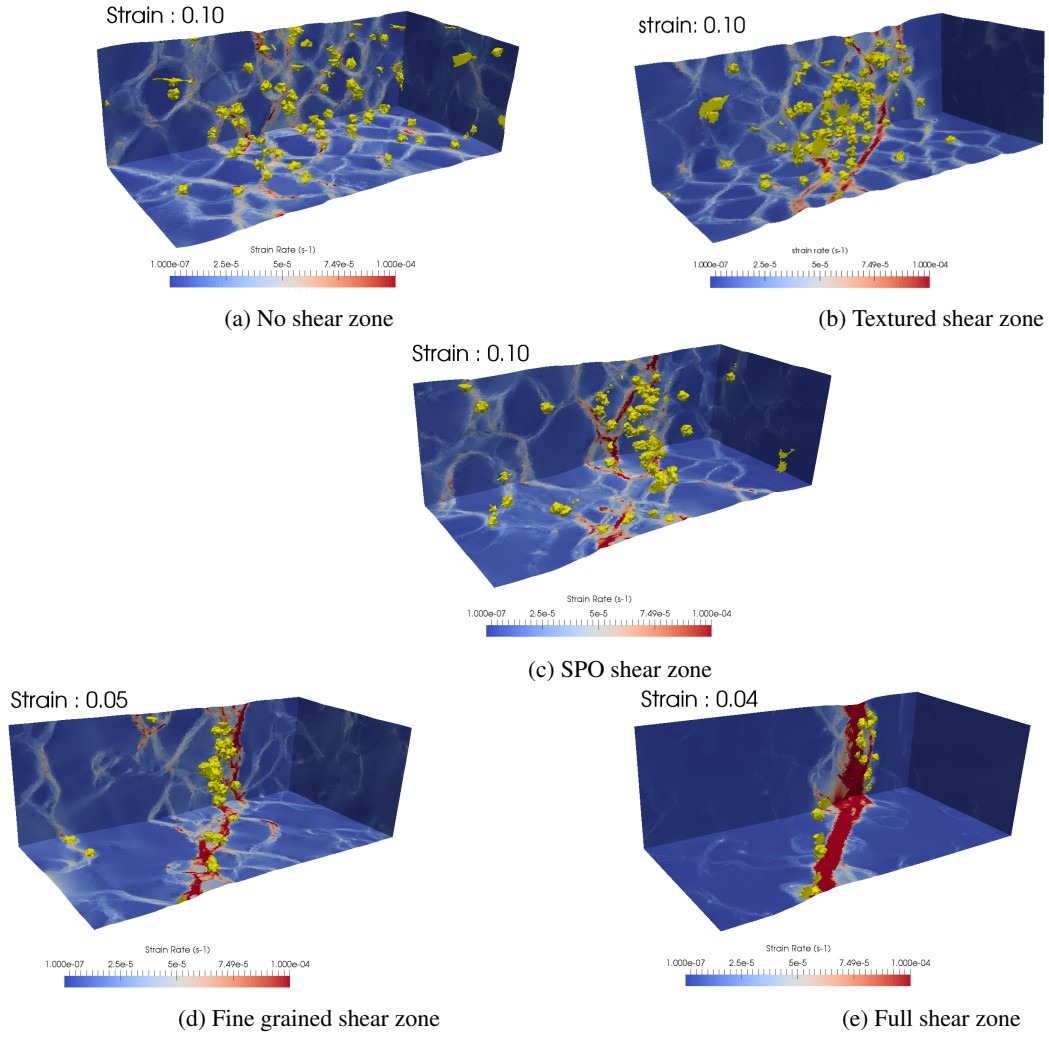

**Figure 7.** Samples after 10% of macroscopic deformation for, no pre-existing 7a, textured 7b, SPO 7c shear zones, after 5% of macroscopic strain for small grain size 7d and after 4% of macroscopic strain for textured plus small grain size shear zone 7e. Colors indicate the strain rate intensity and yellow volumes represent the dynamically recrystallized grains.

zone combining small grain size and crystallographic texture, the strain accumulated by the inner part is almost equivalent to the sum of the two previously described models. The strain accumulated in the outer parts is much lower than for the other

cases (3800% higher in the inner part than in the outer parts after 4% of macroscopic strain, Fig.6). Strain rate distribution within the sample exhibits a large contrast between the inner zone strongly deformed and the outer parts mainly undeformed (Fig.7e). As for other models with a pre-existing shear zone, the nucleation events are located within and near the weakest part of the sample (Fig.7e).



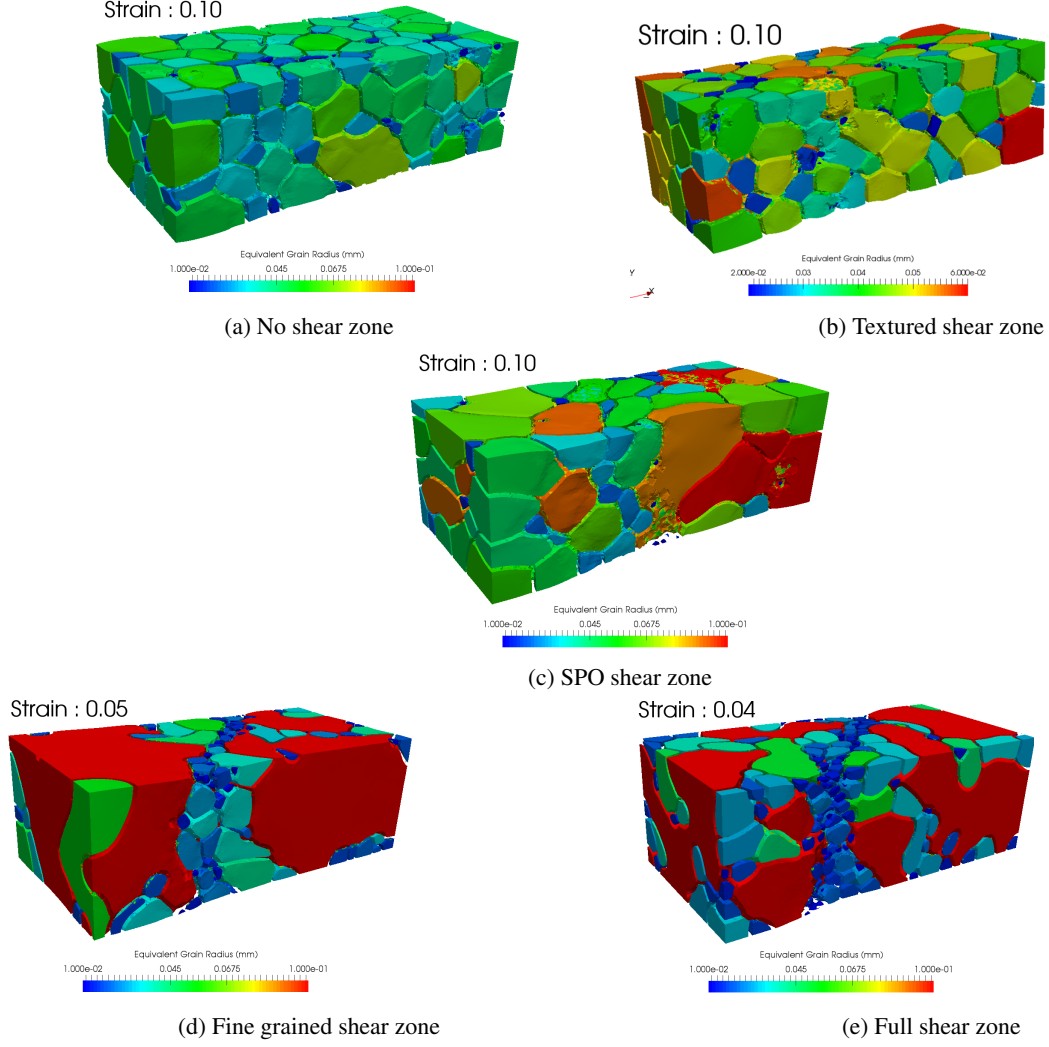

**Figure 8.** 3D view of the microstructure (colors represent equivalent grain radius) for, no pre-existing 8a, textured 8b, SPO shear zones 8c after 10% of macroscopic deformation, for small grain size 8d after 5% of macroscopic deformation and for textured plus small grain size shear zone 8e after 4% of macroscopic deformation.

Polycrystals also undergo grain growth and/or grain size reduction during extension test (Fig.8). For textured and no shear

zone models, the mean grain size is approximately constant during the deformation (Figs.8a and 8b). For small grain size and "full" shear zone models the mean grain size evolutions are similar : large grains of the outer parts are consuming small grains of the inner part (Figs.8d and 8e). For the SPO model, the grain sizes within the outer parts are nearly constant while the grain of the inner part, initially elongated are becoming more equiaxed (Fig.8c).





## 5 Discussion

In the different full field models presented in this study, the development of a strain-localizing band and its characteristics strongly depend on the nature of the pre-existing weak zone. All models exhibit important strain localization, excepted the one with no initial weak zone. In this model, the interior of the aggregate deforms more than its outer parts, but the strain is mainly distributed along GB without exhibiting a clear strain localizing band at the sample scale. Although spontaneous development of a weak band could have been expected as a consequence of grain size reduction through DRX, for instance

near grain undergoing strong dislocation glide deformation, it has not been observed in our model. This can be due to the high temperature considered in this model allowing an efficient healing through grain growth. Moreover, the low final strain of this traction simulation may also explain the non-formation of a shear band. Finally, it could possible that the shear heating (not considered in this work) plays a crucial role in the shear band formation, even at the microscale.

In the four models with a pre-existing weak zone, a strain localizing band develops during the deformation of the sample.

Depending on the nature of the initial weak zone, the mechanical process at stake for strain localization is different. The crystallographic textured shear zone localizes strain through the plastic anisotropy of olivine, due to the lack of slip systems and the heterogeneity of CRSS. This importance of dislocation glide in the textured shear zone is highlighted by several nucleation events occurring within and near the shear zone.

The SPO and small grain size shear zone models are expected to localize strain by increasing the total GB volume within

the shear zone. The heterogeneity of the relaxation mechanism strength between GB and grain interiors is the main parameter controlling strain localization in these cases. The intensity of strain localization seems to be controlled by the grain boundary density within the shear zone. In fact, the grain boundary volume within the shear zone in SPO model is lower than in the small grain size model (for which the strain localization is higher). The two parameters permitting to localize strain (i.e. dislocation glide anisotropy and of the relaxation mechanism strength) seem to have comparable impacts on strain localization, as the

magnitude of strain accumulated within the shear zone is similar for the three models (i.e. textured, small grain size and SPO).

The "full" shear zone model, which is characterized by small and well-oriented grains, shows that the two previous mechanical processes can combine, resulting in a much stronger strain localization than when they act separately. This could be expected from our numerical formalism, as the introduction of the relaxation mechanism does not inhibit the contribution of dislocation glide to the deformation. However, the plastic anisotropy due to the lack of slip system is severely reduced by

the introduction of this relaxation mechanism, as long as its strength is small (such as within grain boundary). As the strain localization in this model seems to sum up the contributions of both mechanisms we can infer that the plastic anisotropy due to the lack of slip systems is small compared to the one due to the heterogeneity of CRSS.
In all models with an initial weak zone, nucleation events are mainly located inside and near the inner part of the sample. From this observation, one can expect that mean grain size within the shear zone should decrease, and for textured and SPO

models it should eventually lead to the formation of a fine grain band. However, for the latter models and for the model without pre-existing shear zone, the mean grain size remains almost constant during the deformation: some of the newly created nuclei rapidly shrink and disappear, which makes it impossible to define a fine-grained zone. For the two other models, even if nucleation events are also dominantly located inside the shear band, the mean grain size quickly increases as grains within the shear zone undergo rapid grain growth. This can be explained by the high dislocation densities within those grains due to

their strong deformation. As a consequence, grains of the outer parts, which are much less deformed (implying a high driving pressure for GBM from difference in stored energy), are consuming the grains of the inner part. Thus, our models cannot predict neither the development nor the persistence of a fine grain zone (mylonite or ultramylonite-like). A few reasons can explain this observation.

First, the nuclei shrinking can be linked to a disequilibrium between the driving forces for GBM: indeed, the capillarity force

tends to shrink nuclei while difference in stored energy leads to their growth. The shrinking of nuclei is then characteristic of a too small difference of stored energy. On this point our constitutive model is tainted of uncertainties because the dislocation density levels are strongly related to the Taylor coefficient (see section 2.1.2). In fact, for a same CRSS value, multiple $(K_1, K_2, \psi)$ triplets are possible but involve different dislocation density levels. For olivine, the value of the Taylor coefficient from the state of the art ranges from $0.5$ to $2.8$ Durinck (2005) which results in very different dislocation density levels. The value

chosen for this study could thus be too high and gives too low dislocation densities which results in a rapid shrinking of nuclei. A second possible explanation could be that the thermomechanical conditions used for these models do not permit to develop and/or maintain a fine-grained zone. In fact, the typical microstructures representative of high temperature deformations do not exhibit fine grain bands, which are much more characteristic of intermediate or low temperature microstuctural features Braun et al. (1999). The last explanation could be tested using our full field frameworks at lower temperatures. Nevertheless, the

material parameters and particularly the nucleation rate $K_g$ is known to be strongly sensitive to temperature. Its dependency with temperature should then be investigated using a set of experimental data before being setup in numerical experiments.

## 6   Conclusions

In this work, we have presented a full field approach for the modelling of deformation and DRX within olivine aggregates. The mechanical framework, based on a reduced CP formulation adapted for the specificities of olivine is able to capture creep

regimes controlled by both grain boundaries (e.g. GBS) and grain interiors (dislocation creep). The LS approach coupled with

the CP framework permits to model microstructural evolutions such as GBM and DRX.

Using this framework, the interdependence between the microstructural features and the mechanical behavior of olivine aggregates has been highlighted. By studying strain localization at the polycrystal scale on pre-existing shear zones, we have

shown that the formation, the intensity and the shape of the developed strain localizing zones are dependent on the nature

of the pre-existing shear band. Particularly, our models show that the intensity of strain localization triggered by a crystallographic textured or a small grain size shear zones are nearly equivalent while these intensities are added when the shear zone is

composed by both textured and small grains. These results underline the importance of taking into account the microstructural

characteristics of crystalline aggregates in the rheology of the lithosphere, and not only the mean grain size or LPO, as these

two characteristics affect the mechanical behavior with the same intensity and cumulate with each other.

*Code and data availability.* The data for supporting all figures of the paper are avalaible upon request to the authors. Moreover, the authors

are interested in scientific collaborations with readers who would like to use the numerical framework presented here.

*Author contributions.* JF, CP, CG, MB and DPM designed the project. JF and DPM designed the numerical framework. JF, CP, CG, MB and

DPM were part of the discussion and contributed to the writing of the manuscript.

*Competing interests.* The authors declare that they have no conflict of interest.

*Acknowledgements.* This work was supported by CNRS INSU 2018-programme TelluS-SYSTER.

The support of the French Agence Nationale de la Recherche (ANR), ArcelorMittal, FRAMATOME, ASCOMETAL, AUBERT&DUVAL,

CEA, SAFRAN through the DIGIMU Industrial Chair and consortium are gratefully acknowledged.



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
