# Peer review of "A new finite element approach to model microscale strain localization within olivine aggregates"

_Solid Earth, 2021_

## Author Response (AR1)

First we wanted to thank the reviewer 1, Thomas Poulet for his careful reading of our paper. His review shows an accurate analysis of the ins and outs as well as of the weaknesses of our work. We mostly agree with the remarks he provides. Second, we deeply apologize for having used the wrong reference format; since we were focusing on the scientific content of this paper, we missed this issue at the moment of the submission. Apparently, it is a major flaw that should justify rejection of the paper, because it alters so much its substance that it makes it unreadable for reviewer 2 – we are surprised. Fortunately, this problem is not very difficult to fix, so we will correct the reference format and at the same time we will focus on replying to referee 1's remarks.

The main points underlined by the first referee concern the calibration of the model parameters, which is in fact an arduous task and could be considered as intricate as the development of the numerical framework itself. Indeed, for such a model to be predictive in a geologically relevant context, numerous parameters have to be identified for strain rate and temperature conditions largely out of the range of experimental conditions. The aim of the present article was neither to pretend applying directly this framework to geological contexts, nor to cover the entire spectrum of laboratory experiment conditions. Instead, this work presents a first, decisive step toward this objective, which is to present the numerical methods with their underlying physical concepts, their adaptation to mantle rocks, and illustrate its capability through seemingly "straightforward" results. We understand the disappointment of the reviewer about the conclusions presented in the paper, but we are convinced that these results will prove the reliability of the framework which is a key step in order to tackle more fundamental questions in the near future. We will try in the following responses and in the revised manuscript to answer the legitimate remarks of the referee and enlighten the choices made for this work, especially for parameter identification.

The parameter identification is expensive in terms of computational resources particularly for this type of 3D full field simulations with FE remeshing algorithms (increasing CPU time). Indeed, some of the calibrated parameters are strongly interconnected, as the recrystallization rate $K_g$ with the LAGB mobility $M_{LAGB}$, and the average strain accumulated above which a sub-grain can be considered as a grain $\varepsilon^{cum}_{HAGB}$, which implies to test an important number of combinations of these values. This is a reason why, among the 22 parameters presented in table 1, we considered just a few of them as fitting parameters (all presented in the section 3). Moreover, the majority of the presented parameters in this paper are often quantified in the olivine-related literature, and we deliberately fixed them according to these studies for two reasons: first, some material parameters as critical resolved shear stress (CRSS) or Burger's vector are not supposed to vary between models as they represent an intrinsic property of the material. Second, we want to avoid any overfitting

temptation, because we think that any model with 22 degrees of freedom could fit any arbitrary dataset, without necessarily being meaningful. Instead, our approach tries to account for physical mechanisms impacting deformation and microstructural evolutions and the discrepancies observed with some experimental dataset may just mean that some mechanisms are not well described by our model, or/and that we have missed some of these mechanisms, or/and the experimental procedure could be marred by uncontrolled factors. Still, we prefer to rely on this imperfect fit rather than on other poorly constrained parameters.

As an example illustrating the above remark, referee #1 has been particularly circumspect about the calibration of hardening and recovery parameters K1 and K2 values, and about the gaps between simulated and experimental data. Indeed, for low strains, the strain/stress numerical curves deviate from experimental ones. This first stage of the sample mechanical behavior is related to the activation of crystal slip systems when the resolved shear stress reaches the CRSS. The model overestimation of the stress around this point could mean that the CRSS values used are too high. However, we have chosen to fix the values of CRSS because they correspond to commonly reported values in numerous experimental studies. We use for them an analytical expression derived from dislocation dynamics embracing a large experimental dataset and giving us, in addition, a temperature dependence law for these material parameters. The discrepancies observed between experimental and modeled strain/stress behavior may thus be explained by other mechanisms not considered in our model (low temperature mechanisms such as twinning), or badly described by our strain accommodating mechanisms (currently considering an isotropic relaxation of the stress). Another source of error, now discussed in the manuscript, may also come from experimental study used here (Phakey et al. 1972). Indeed, Phackey et al. (1972) mentioned that fractures have been observed in their samples, which can reflect a lower elastic limit than commonly assumed (thus explaining the large gaps between model and experiments at low strain). Moreover, the [110]c 1073K setup, which is the less compatible with our model predictions has been qualified by Phakey et al. as highly stressed, which is consistent with the underestimation of the stress by our model.

Concerning the temperature effect on the various components of the model, numerous material parameters already contains an explicit temperature dependency. These parameters are, for the crystal plasticity (CP): the CRSS, K1 and K2, for the reduced CP: the strength of the relaxation mechanism and for the microstructural evolutions : the grain boundary mobility, the solute drag parameters and the critical dislocation density. An important lack in temperature dependency parameters is the nucleation rate, for which we did not find any relevant experimental dataset in the literature. A next step for including temperature effects in this model could be to consider heat generation due to plastic flow dissipation.

As mentioned in the above paragraphs and now explained in the manuscript, we are aware that our framework is still suffering from a lack of parameter calibration for a vast spectrum of thermomechanical conditions. For this purpose, it could be really useful if all olivine polycrystal deformation experiments could report mean grain size and recrystallized fraction evolution during deformation, in order for us to identify the recrystallization rate for different thermomechanical conditions. Nevertheless, we think that our manuscript could be of interest for Solid Earth readers because it presents a new full field model for the simulation of olivine aggregates deformation and such a numerical framework –even though it is still perfectible -could be of use for several applications in solid earth sciences.

Concerning the minor comments:
- We agree that we have been a bit restrictive by presenting microstructural features and related processes as the key factor for the initiation of ductile strain localization. We have now opened this discussion to larger-scale processes in the introduction according to the recommendation of the reviewer.
- Concerning the discretization, we have now explained in the revised manuscript that our P1 elements are elements associated with linear interpolation functions.
- About the remeshing, this time-periodic strategy enables to consider large deformation of our representative volume elements (RVEs) without any FE convergence difficulty. Indeed, the idea behind consists of always keep a good quality field for all the finite element of the mesh whatever the considered deformation. Typically, it was proved that such an approach can be used to model dynamic recrystallization in large rolling of metallic materials RVE.
- Regarding Eq.22 defining the equivalent stress to be compared with experimental ones, we have tried to compute the stress in the same way that experimentalists do, dividing the applied force by the section area (this could also be seen as the average stress over the cross section). Thus in order to compute this force we need to integrate the elementary stress - known thanks to the FE resolution - on a surface (as a force could be seen as a 2D integral of stress). Of course the vertical position of this cross cutting surface may seem arbitrarily chosen, but we have verified that it does not impact much the results.
- For Fig.4b, the model predicts stress values very close to the ones expected with the deformation map. In fact, just before the strong grain size reduction at 5% of macroscopic strain, the deviatoric stress equals 190 MPa and in the constant grain size regime (after 10% strain), the predicted deviatoric stress is 130 MPa. We have added this information in the revised version of the manuscript.

- Concerning the weak strain localization in the model without an inherited shear zone: this is due to the geometry and boundary conditions, which leads to a higher deformation in the inner part of the sample. As mentioned by the referee this case serves as a reference test permitting to decorrelate the strain localization due to the geometry and boundary conditions from the one due to microstructural characteristics. These explanations have been added to the revised version of the manuscript and we have also modified the presentation of the data in Fig.6, hoping this will provide a better comprehension of the results.

- We thank the reviewer for the typos mistakes he raised. The notation for the fourth order identity tensor used here is the latex \mathbb{1} symbol. For the boundary conditions presented in line 281, the displacements on the lower face are in fact imposed null in all directions; we have precise it in the revised version.

---

## Author Response (AR2)

Dear SE editor,

We understand the point of view of Thomas Poulet in his last report and we would thank him again for his sincerity and his serious work on this review. As you we trust that the open access nature of SE may allow for the interested community to get access to this work and thus we also want to thank your decision. Following your recommendation, we have added some comments concerning the limitations and the validity of our framework in the discussion of the revised manuscript.

Best regards,

Jean Furstoss on the behalf of the authors.